# The Tumorigenicity of Breast Cancer Cells Is Reduced upon Treatment with Small Extracellular Vesicles Isolated from Heparin Treated Cell Cultures

**DOI:** 10.3390/ijms242115736

**Published:** 2023-10-29

**Authors:** Yunliang Chen, Michael Scully

**Affiliations:** Thrombosis Research Institute, 1b Manresa Road, London SW3 6LR, UK; mscully@tri-london.ac.uk

**Keywords:** breast cancer, heparin, small extracellular vesicles, miRNA, cytology profile

## Abstract

As a member of the HPSG family, heparin is often used as a specific probe of their role in cell physiology; indeed, we have previously shown a reduction in the tumorigenicity of breast cancer cells when cultured in its presence. However, a partial reversal of the anti-tumorigenic effect occurred when the treated cells were cultured in fresh medium without heparin, which led us to consider whether a more persistent effect could be achieved by treatment of the cells with small extracellular vesicles (sEV) from heparin-treated cells. The tumorigenicity was analyzed using sEV isolated from the culture medium of heparin-treated MCF-7 and MDA-MB231 breast cancer cells (sEV-HT) or from conditioned medium following the termination of treatment (heparin discontinued, sEV-HD). Tumorigenicity was reduced in cells cultured in the presence of sEV-HT compared to that of cells cultured in the presence of sEV from untreated cells (sEV-Ctrl). sEV-HD were also observed to exert an anti-tumorigenic effect on the expression of pro-tumorigenic and cell cycle regulatory proteins, as well as signaling activities when added to fresh cultures of MCF-7 and MDA-MB231 cells. The anti-tumorigenic activity of the heparin-derived sEV may arise from observed changes in the miRNA content or from heparin, which was observed to be bound to the sEV. sEV may constitute a relatively stable reservoir of circulating heparin, allowing heparin activity to persist in the circulation even after therapy has been discontinued. These findings can be considered as a special additional pharmacological characteristic of heparin clinical therapy.

## 1. Introduction

Heparan-sulfate proteoglycans (HSPGs) are present throughout the cell surface where they play a fundamental structural role, as well as a modulatory one, controlling the bioavailability and function of a range of autocrine and paracrine signaling molecules (e.g., growth factors and chemokines) [1]. In heparan sulfate (HS), the polysaccharide component is capable of interacting with a multiplicity of proteins, mainly through ionic binding, and thereby controlling key aspects of cell genotype and phenotype, and it is implicated in a wide range of pathophysiological settings including cancer. As members of the glycosaminoglycan family of carbohydrates, heparin and HS have similar polysaccharide structures, consisting of variably sulfated repeating disaccharide units; consequently, heparin is often used as a convenient probe to investigate the role of HPSGs in cell biology [2,3].

It is well known that heparin, including unfractionated heparin (UFH), low molecular weight heparin (LMWH), and heparin derivatives, is used for the prevention and treatment of venous thromboembolism which is often associated with cancer, and in clinical trials it has been established that heparin treatment has beneficial effects on cancer survival [4]. There are indeed a number of reports showing that LMWH can improve the 3-month and 6-month survival of cancer patients [5,6,7,8,9,10]. A multicenter clinical trial exploring the influence of anticoagulant treatment in 277 small-cell lung cancer patients showed that 5-week subcutaneous heparin treatment led to substantially improved survival rates compared to no treatment at 1, 2, and 3 years (40% vs. 30%, 11% vs. 9%, and 9% vs. 6%, respectively) [11].

In our previous study, we observed that culturing of breast cancer cells in the presence of heparin exerted a broadly anti-tumorigenic effect by reducing the expression of pro-tumorigenic proteins, as well as the level of activation of key cell signaling pathways [12]. This anti-tumorigenic effect was partially reversed upon the replacement of fresh medium without heparin, which led us to consider whether a more persistent effect could be achieved by treatment of the cells with sEV produced by the heparin-treated cells, since sEV are known to act as intercellular messengers and are remarkably stable in body fluids.

sEV, named selectively according to the MISEV 2018 guidelines and previously termed exosomes [13], are small extracellular vesicles (30–90 nm) [14,15], produced by a wide variety of cell types including reticulocytes, epithelial cells, neurons, and tumor cells [16]. Because of their bioactive contents (proteins, mRNA, miRNA), sEV constitute concentrated carriers of genetic and proteomic information, which vary in properties and functionality according to the cell type from which they originate, as well as the particular physiological and pathological conditions in existence at the time of their packaging and secretion [17,18]. By shuttling molecules between cells, sEV promote intercellular communication and alteration in the function of target cells and, thereby, are considered to be involved in modulating a wide range of cellular processes. sEV have also been identified in body fluids and blood, indicating that the exchange of information between organs may also occur via sEV [19,20]. 

In the present investigation, we compared the properties of sEV prepared from heparin-treated breast cancer cells (sEV-HT) to those from untreated cells (sEV-Ctrl) as well as to those separated from conditioned medium following the termination of treatment (heparin discontinued, sEV-HD). Heparin treatment was found to generate a less tumorigenic genotype/phenotype in the sEV which was transferred when the heparin-treated sEV (sEV-HT) were added to fresh cultures of cancer cells. A similar transferable modulatory effect was also found in cancer cells treated with sEV-HD, sEV produced by cells, 2–3 days after halting heparin treatment.

## 2. Results

### 2.1. Antitumorigenic Effects Observed in Heparin-Treated Human Breast Cancer Cells Persist Temporarily after Heparin Has Been Discontinued

Our previous work established that heparin treatment is able to reduce the pro-tumorigenic profile of breast cancer cells according to a number of parameters including the reduction in the level of activation of the PI3K/Akt and MAPK/ERK signaling pathways [12]. To determine whether the anti-tumorigenic effect of heparin was retained after heparin treatment was discontinued, MCF-7 cells were treated with UFH (10 IU/mL) for two days in culture medium containing 0.2% of LAH, at which point, the medium was replaced with fresh medium without heparin and cultures maintained and sampled at 1, 2, or 3 days. Untreated cells were used as a control. Cell proliferation, cell cycle, and Western blot assays were carried out. The cell proliferation rate was reduced in the heparin-treated group and this reduction was maintained after the removal of heparin as observed on Days 1 and 2 (Figure 1A). Cell-cycle profiling showed that treated MCF-7 cells were growth-arrested in the G0-G1 phase together with a concomitant reduction in S-phase. These changes in the cell cycle profile in heparin-treated cells were also observed on Days 1 and 2 after heparin was withdrawn from the culture medium (Figure 1BI, II). Western blots demonstrated that heparin modulated levels of p-ERK and p-Akt, but this effect was only observed on Day 1; however, the heparin-induced reduction in the expression of the cell cycle regulatory proteins CDK4 and cyclin-D3 persisted for up 3 days after heparin treatment had been discontinued (Figure 2). 

### 2.2. Similar Changes Are Observed in the Expression Profiles of Apoptotic and Pro-Tumorigenic Proteins following Heparin Treatment in the Cells and in the sEV Produced by the Cells (sEV-HT)

To find out whether heparin is able to modulate the levels of apoptotic and pro-tumorigenic proteins in both sEV and their host cells, the expression of a number of specific examples was assayed by Western blot in sEV-HT separated from the conditioned medium of heparin-treated MCF-7 and MDA-MB231 cells. In comparison to sEV separated from untreated cells, an enhanced expression of active caspase 3, pro-apoptotic caspase adaptor protein, and TNF-α was observed, as well as a reduced expression of mucin 1 (MUC1), (one of the most highly upregulated proteins in multiple tumor types) in sEV-HT isolated from both UFH- and LMWH-treated MCF-7 and MDA-MB231 cell cultures. These modulatory effects were similar to those observed in cell lysates from heparin-treated MCF-7 cells (Figure 3). 

### 2.3. Changes Observed in the miRNA Profile of Heparin-Treated Host Cells, Which Are Pertinent to Tumorigenesis and Cell Cycle Regulation, Are Also Observed in the sEV Released by the Cells (sEV-HT) 

We first determined whether heparin treatment of breast cancer cells was able to modulate the miRNAs profile, since this is considered to contribute to the fine tuning of gene expression at a global level. Based on our previous work, several miRNAs were selected which are known to be involved in the regulation of heparin-modulated pro-tumorigenic genes, such as MUC1, Akt [21,22]; and cell cycle regulatory proteins [21,22,23,24,25,26,27]. Heparin treatment of MCF-7 cells was shown to modulate the expression of each of the miRNAs selected for testing; in particular, the level of miR-155 was reduced, whereas the levels of miR-10, miR-21, miR23a, miR137, miR125b1, and miR-145 were increased (Figure 4A). A similar modulation of the miRNA profile was also observed in RNA samples isolated from the sEV produced by heparin-treated MCF-7 cells (sEV-HT) (Figure 4B).

### 2.4. The miRNA Profile of Breast Cancer Cells Is Modulated following Treatment with sEV-HT and sEV-HD

Since similar expression profiles of pro-tumorigenic proteins and miRNA were observed in sEV-HT and their heparin-treated host cells, it was posited that the effect of heparin may be transferable, in the absence of heparin, to other tumor cells by the genetic and proteomic information contained within the heparin-derived sEV. It is known that sEV contain inactive forms of both mRNA and miRNA that can be transferred by uptake of the sEV into target cells and act in a functional role [28], thereby playing an important role in intercellular communication.

To test this hypothesis, MCF-7 and MDA-MB231 cells were each treated with their respective exosome preparations, sEV-HT or sEV-HD, as well as with a control preparation of sEV separated from untreated cells, sEV-Ctrl. Analysis demonstrated that in comparison to sEV-Ctrl-treated cells, there was a significant increase in the level of miRNA 10b, miRNA21, miRNA23a, miRNA125b-1, miRNA137, and miRNA145 in both cell lines, as well as a reduction in the level of miRNA155. These alterations in the miRNA expression profile were similar in comparing the effect of sEV-HT treatment to that of sEV-HD in MCF-7 and MDA-MB231 cells (Figure 4B and Figure 5A, respectively). It is concluded that the effect of heparin treatment on the miRNA content of sEV is transferable to other cells via sEV and is sustained since the modulatory effect occurs in the absence of heparin. 

### 2.5. Pro-Tumorigenic and Cell Cycle Regulatory Proteins as Well as Signalling Activities Are Modulated in Breast Cancer Cells Treated with sEV-HT and sEV-HD

MCF-7 and MDA-MBA231 cells were treated with each of the two types of sEV preparations (sEV-HT and sEV-HD) for 24 h following which the level of protein expression and signaling activities of the treated cells were analyzed by Western blot assay. In comparison to the effect of treatment with control sEV prepared from cells cultured in the absence of heparin, we also observed that the level of p-Akt in MCF-7 and MDA-MB231 cells was reduced by treatment with both preparations of the heparin-derived sEV (Figure 6A). The level of p-ERK activity was found to be reduced in MCF-7 cells treated with sEV-HT but was unchanged upon treatment with sEV-HD. In MDA-MB231 cells, the level of p-ERK activity was reduced following treatment with LMWH. An enhanced level of p-p53 was found upon MDA-MBA231 and MCF-7 cells’ treatment with both sEV-HT and sEV-HD. Enhanced expression of p21 was observed in sEV-HT-treated MCF-7 and MDA-MBA231 cells, and SEV-HD treatment also affected MDA-MB231 cells (Figure 6A). Each of these changes was similar to that observed when cells were directly treated with UFH or LMWH. 

It is well known that both p53 and p21 act as important tumor suppressors in a cooperative manner and p53-p21-RB (retinoblastoma protein) signaling contributes significantly to cell cycle regulation [29]. Thus, compared to exosome-treated controls, the expression and activities of the tested cell cycle regulatory proteins in the two cancer cell lines (MCF-7 and MDA-MBA231) were found to be reduced by the treatment with both preparations of sEV (Figure 6B). The degree of modulation observed was similar, in each case, to that observed upon direct treatment of the cell line with UFH or LMWH (Figure 6B). These results show that the heparin-induced effect on the level of expression of pro-tumorigenic protein and cell cycle regulatory proteins, as well as signaling activities, can persist and be transferred when targeted tumor cells are treated with heparin-treatment-derived sEV. 

### 2.6. The Cytological Profile Is Also Modulated upon Treatment of MCF-7 and MDA-MB231 Cells with SEV-HT and SEV-HD

To determine whether the cytological profile of breast cancer cells could also be affected by treatment with sEV derived from heparin-treated cells, MCF-7 and MDA-MB231 cells were cultured with different preparations of sEV, which included heparin-untreated sEV as a control and both UFH- and LMWH-treatment-derived sEV-HT and sEV-HD. After 24 h, cell cycle and apoptosis analysis were carried out by flow cytometry. Using non-sEV-treated cells as a reference, the treatment of MCF-7 and MDA-MB231 cells with heparin-derived sEV was observed to exert a similar effect on the cell cycle profile as the treatment of cells with heparin alone. The S and G2-M phase were significantly reduced in the heparin-derived-exosome-treated groups in a similar way to that observed previously upon the direct treatment of the cells with heparin, the only marked difference being the slightly greater reduction in the S phase upon direct treatment. The somewhat lesser effect on the S phase observed in the exosome-treated cells may be caused by a reservoir of growth factors and cytokines which exists in sEV (Figure 7A,B). The increase in apoptotic rate observed upon direct treatment of MCF-7 and MDA-MB231 cells was also observed when cells were treated with heparin-derived sEV (Figure 8).

### 2.7. Heparin Binds to the Surface of the sEV

Using heparin-FITC (Thermo Fisher Scientific, USA) and flow cytometry, we observed that heparin is liable to uptake by each of a number of cancer cell lines (Appendix A). These observations were confirmed by confocal microscopy (Leica DMI4000B, German), which demonstrated that FITC-heparin co-localized with Dil, a marker for endoplasmic reticulum (ER) and Golgi regions, indicating that heparin closely associates with the nucleus (Appendix A). As a highly sulfated glycosaminoglycan, heparin is able to bind a variety of protein and molecular ligands [30]. To find out whether heparin is able to bind to the surface of the sEV cargo which is comprised of a variety of protein and molecular entities, the sEV separated from FITC-heparin (1 µg/mL) treated MCF-7 cells were aliquoted into microwell plates and analyzed using a Molecular Devices Fluorescence Plate Reader. The results demonstrated that FITC-heparin bound to sEV and the binding was competitively inhibited by increasing concentrations of UFH as determined from 0 to 2 IU/mL (Appendix A). Flow cytometric analysis of FITC-heparin-treated sEV showed binding to Exo-Flow FACS magnetic beads pre-coated with sEV surface marker antibodies, CD9 or CD63, and the binding of heparin was shown to be readily reversible by the addition of UFH (Figure 9).

The results above show that heparin binding to sEV as well as heparin-loaded sEV could act as a longer-lived reservoir of heparin activity in the circulation after heparin administration has been terminated. These observations provide a mechanism by which the direct anti-tumorigenic action of heparin on the cancer cells we reported previously could be transferred via heparin-derived sEV and thereby persist in the circulation long after heparin treatment has been discontinued. 

## 3. Discussion

Since its first discovery, a range of explanations have been proffered to explain the beneficial effects of heparin treatment upon the survival of cancer patients at risk or suffering with thrombosis [30]. To date, a number of preclinical and clinical studies have established that the effect of heparin is independent of its antithrombotic effect and is more likely to be linked to its ability to modulate the activity of many of the bioactive agents and pathways which are controlled, at least in part, by interaction with cell surface HSPGs [12,30,31,32]. Heparin is derived from a highly sulfated member of the HSPGs family and can bind to a wide range of proteins and molecules via electrostatic interactions with the polyanionic groups of the sulfated glycosaminoglycan chains. This group of proteins and peptides, termed the heparin/heparan sulfate interactome, includes a wide variety of crucial growth factors and other proteins [33].

As micro-vesicles, sEV are ubiquitously released into the extracellular milieu and have been identified in many biological fluids and tissue culture media. It has been suggested that sEV may contain cell-specific factors that allow the sEV to target specific cells for the delivery of the sEV contents (proteins, mRNAs, miRNAs, and DNAs which reflect the originating host cell) to recipient cells [34]. sEV interact with recipient cells, which may be local or considerably distant from the original cell surface, through a process entailing ligand/receptor signaling at the recipient cell surface or the fusion of vesicle and cell plasma membranes [17,18]. It has been shown that through an interaction with a range of host tissues, the tumor-derived sEV are able to generate a pro-tumor environment that is essential for carcinogenesis [35,36]. This present work is the first to show that sEV isolated from heparin-treated cancer cells are able to induce a modulation of pro-tumorigenic characters including alteration in the levels of tumorigenic proteins and signaling pathways, with consequential effects upon apoptotic and cell cycle regulation in a similar way to the effect of the direct treatment of the breast cancer cells with heparin which we have reported previously.

sEV are also capable of transferring their cargo of nucleic acids to the recipient cells, and thereby affecting protein production and inducing phenotypic changes [35]. Tumor-derived sEV have been shown to contain a wide range of nucleic acids, with most studies investigating exosome miRNA or messenger RNA (mRNA), which regulate the stability or translational efficiency of targeted messenger RNAs [36]. Their aberrant expression is thought to be involved in a range of human diseases, including cancer. For example, miR-10b, miR-125b, and miR-145 have been shown to be downregulated in human breast cancer cells, while miR-21 and miR-155 were upregulated, from which observations it has been proposed that these miRNAs may act as potential tumor-suppressor genes or oncogenes [37,38,39,40,41,42]. A number of studies have also demonstrated that miRNAs in sEV can influence target cell function [16,36,37,43]. As master regulators, individual miRNAs could be associated with numerous target genes and be involved in a wide array of cellular processes. miR-21 and miR-137 were both found to be associated with the PI3K/AKT signaling pathway, and are thereby involved, in the case of miR-21, in tumor angiogenesis [22] and in the case of miR-137 in tumor repression [37]. miR-145-mediated suppression of cell invasion has been shown to be due, at least in in part, to the silencing of the metastasis gene, mucin 1 (MUC1) [21]. miR-125b, which is downregulated in breast cancer cells, also suppresses translation of the MUC1 oncoprotein [38], while miR-23a is considered to be involved in TNF-α-induced endothelial cell apoptosis [44]. Many of the oncogenic or tumor-suppressor miRNAs also target cell cycle regulators and impact cell proliferation. miR-125 regulates the proliferation of cancer cells through the inhibition of the expression of cell cycle regulatory proteins, CDK6 and Cyclin A2 [23,24]. miR-137 inhibits the proliferation of lung cancer cells by targeting Cdc42 and Cdk6 [31], while miR-145 can cause cell cycle arrest at the G0-G1 phase, decrease the S-phase, and inhibit proliferation [27]. miR-137 in lung cancer cells significantly downregulates Cdc42 and Cdk6 and induces G1 cell cycle arrest, leading to a significant decrease in cell growth [26]. Our work found the levels of the deregulatory miR-10b, miR-125b, miR-145, and miR137 were enhanced by the addition of heparin or heparin-treated sEV to cultured cells with observed effects on the level of their putative targets; for example, a reduction in p-Akt, MUC1, and a number of cell cycle regulatory proteins. These effects are accompanied by a concomitant change in cell cycle profiles observed as an arrested G0-G1 phase and decreased S-phase, as well as reduced cell proliferation activity. miR-155 is a typical multifunctional miRNA, which has been found to be over expressed in several solid tumors, including breast cancer, where it is involved in promoting cancer cell differentiation [36,39,40]. It is considered that the estimation of levels of miRNA-155 in tissues or biological fluids may be useful for tumor detection and for evaluating prognosis [41,42]. In this respect, our finding that the level of miR-155 was reduced by heparin treatment or in sEV from heparin-treated cells may be highly significant in considering the anticancer activity of heparin’s effect on human breast cancer.

In addition, we also observed another feature of sEV behavior that may help to explain the long-term effect of heparin, namely the uptake of heparin by the sEV. Several laboratories have shown previously that heparin is liable to intracellular uptake when added to cells [44,45,46]. Internalized heparin likely interferes with transcription factor function and subsequently induces apoptotic cell death. Therefore, internalized heparin is considered as a novel mechanism for inducing apoptosis of cancer cells [47]. As a highly sulfated polysaccharide, heparin has also been shown to bind proteins on the sEV surface, thereby blocking a key mechanism responsible for the binding and uptake of sEV by cancer cells through interaction with cell-surface HPSGs [48]. A recent report has also shown an interaction between sEV and heparin and the heparin affinity chromatography (HAC) was used to capture extracellular vesicles [49]. In this present investigation, we observed that heparin is liable to uptake by each of a number of cancer cell lines and is also able to bind to the surface of the sEV separated from heparin-treated MCF-7 cells. The therapeutic dosage of heparin used routinely in cancer patients is 150–300 units UFH or 200 units LMWH per kilogram of body weight administered once daily by subcutaneous injection [50]. In this study, our work has shown that the maximum binding of FITC-heparin by the cancer cells occurs at 1 µg/mL equivalent to around 0.4 unit heparin/mL, similar to the levels observed in blood during heparin therapy, a concentration which will be efficacious for internalization by cancer cells. However, we have yet to investigate whether the observed changes in protein expression occur on the surface of sEV or within and further work is planned.

The anti-tumorigenic effect of heparin treatment of cancer cells was observed upon treatment with heparin-derived sEV. sEV are remarkably stable in bodily fluids and able to interact with the surrounding tissues in both intercellular and extracellular domains; the sEV heparin may also constitute a relatively stable reservoir of circulating heparin, allowing heparin activity to persist in the circulation even after therapy has been discontinued. The significance of our findings in terms of the clinical use of heparin during cancer treatment is that it provides an additional mechanism by which heparin and heparin-like molecules are able to exert a favorable effect on the long-term prognosis of cancer patients. Heparin was first introduced into the treatment of cancer because of the higher incidence of deep vein thrombosis and venous thromboembolism in cancer patients [51]. In addition to the prevention of thrombosis, a variety of heparin-like compounds have been shown to exert a favorable effect upon cancer outcome through the modulation of the interaction of effector molecules (for example, fibroblast growth factor) in driving the growth and development of cancer cells [52]. Our findings are the first to establish that heparin can modulate the progression of cancer via the intracellular messenger role of sEV released by cancer cells turning their role to an antitumorigenic one, rather than a protumorigenic one. This insight will need to be established further within clinical studies in which the genotype/phenotype of circulating extracellular vesicles will be analyzed in patients in relation to the use of heparin or heparin-like mimetics. In this role, the use of the non-anticoagulant heparin may also be effective.

## 4. Material and Methods

Cell Culture: Human breast cancer cell lines, MCF-7 and MDA-MB-231, were obtained from the Health Protection Agency culture collection (UK). MCF-7 cells were maintained in DMEM medium supplemented with 15% fetal bovine serum (FBS, Thermo Fisher Scientific, USA), 100 U per ml penicillin, and 100 mg per ml streptomycin, and cultured in a humidified atmosphere of 5% CO_2_ in air. MDA-MB-231 cells were cultured in Leibovitz’s L-15 medium with the above same supplements and conditions but without CO_2_. 

Western Blot: The cells were cultured until 90% confluence in medium containing 15% FBS, which was then replaced with serum-free medium containing 0.2% Lactalbumin Hydrolysate (LAH) overnight. The cells were subsequently treated with 1–10 IU/mL, unfractionated heparin (UFH, from porcine intestine, Sigma, USA), or low-molecular-weight heparin (LMWH, Fragmin, Pfizer) (as indicated). Harvested cells were treated by lysis buffer (8 M urea, 5%SDS, 40 mM Tris-HCl pH 6.8, and 1 x Protease Inhibitor Cocktail tablet cOmplete (Roche, Germany)). In total, 5 μg of protein per well for each sample was used to run an SDS-PAGE gel, and trans-blot with the Trans-Blot turbo transfer system (BIO-RAD, UK), then the blot was analyzed by iBright-CL1000 (Invitrogen, USA). Primary antibodies towards MUC-1, TNFα, active Caspase 3, and pro-apoptotic Caspase adaptor protein were obtained from Abcam (UK). Primary antibodies towards ERK, Akt, Cell cycle regulation antibody sampler kits I and II, and the appropriate HRP-conjugated secondary antibody were obtained from Cell Signaling (UK). Densitometry was performed using iBright Analysis Software within iBright-CL1000 (Invitrogen, USA) 

sEV Isolation: Secreted sEV were prepared from cell culture medium by differential centrifugation, essentially, as described previously [20]. Briefly, sub-confluent cells were cultured overnight in serum-free DMEM supplemented with 0.2% LAH, 2 mM L-glutamine, 100 U/mL penicillin, and 100 μg/mL streptomycin, which was then replaced with medium with or without UFH/LMWH (10 IU/mL). After 2 days, the culture medium was harvested to prepare the heparin-treated sEV fraction (sEV-HT). The heparin-treated cells were washed with heparin free medium and after the addition of fresh, heparin free medium was cultured for a further 2 days, at which point the culture medium was harvested to prepare the heparin discontinued sEV fraction (SEV-HD). To purify the sEV fraction, the conditioned medium was firstly centrifuged at 300× *g* at 4 °C for 5 min to eliminate cell debris, and then 10,000× *g* for 20 min at 4 °C to remove microvesicles. sEV were then pelleted by ultracentrifugation (Beckman XL90 Ultracentrifuge XL90, rotor type 70 TI) at 100,000× *g* at 4 °C for 2 h and washed with PBS solution followed by ultracentrifugation at 100,000× *g* for 2 h in a Beckmann XL-90 ultracentrifuge. The fresh sEV obtained by ultracentrifuge isolation were used for further analysis. Exo-ExoQuick precipitation kits (System Biosciences, CA, USA) were used in experiments determining whether FITC-heparin (1 µg/mL) was capable of binding to the surface of sEV separated from the conditioned medium of heparin-treated MCF-7 cells. According to the manufacturer’s protocol, Exo-Flow FACS magnetic beads which are pre-coated with sEV surface marker antibodies (CD9 or CD63) were used during flow cytometric analysis in a Cytomics FC500 (Beckman Coulter, USA). In some experiments, in order to corroborate the quality of the ultracentrifugation isolated sVE, an isolation kit (ExoQuick-TCTM, System Bioscience, UK) was used for the purpose of comparison only. The sEV from untreated cell was used as control. Isolated sEV were also counted and assayed using qNano Gold (IZON Science Europe, UK), which showed that 85% of them were between 50 and 150 nm, and 95% of them were between 50 and 500 nm. Based on the assay of the number count and concentration of sEV, breast cancer cells were treated with by 5–10 µg of sEV protein concentration/mL or 4 × 10^9^ sEV/mL in the following work.

Breast cancer cells (MCF-7 and MDA-MB231) were each treated with the separated sEV within culture medium that contains 1% exosome-depleted FBS (System Biosciences, CA, USA) in 6-well culture plates for 24 h. The sEV were removed from the treated cells by washing the cells three times with PBS, then the treated cells were subjected to Western blot and miRNA analysis. 

miRNA Isolation and assay: miRNA from treated cells was isolated using an *mir*Vana^TM^ miRNA Isolation Kit (Life Technologies, UK) according to the manufacturer’s instructions. A TaqMan^®^ MicroRNA Reverse Transcription Kit was used with the Reverse Transcription (RT) primer provided with the specific TaqMan MicroRNA Assay to convert miRNA to cDNA. The cDNA samples were quantitated by TaqMan miRNA assays (Applied Biosystems, UK). Signals were detected using the ABI Prism 7900 HT sequence detector (Perkin-Elmer-Cetus, Vaudreuil, QC, Canada). Triplicate samples were run, and expression values were standardized to values obtained with an endogenous control using miR-RUN6B primers as previously described [12,43]. Statistical analysis was performed using the Student’s paired t- test. 

Proliferation, cell cycle and apoptosis assays: The cells were treated with heparin (5–10 IU/mL) for 1–2 days in FCS-free medium. A proliferation assay was carried out with a CyQUANT^®^ Cell Proliferation Assay Kit (Life Technologies, UK) according to the manufacturer’s instructions. For cell cycle assays, the treated cell suspension was lysed and permeabilized with DNA Prep LPR (Beckman Coulter, USA), the DNA stained with propidium iodide (PI) and analysis carried out in a Cytomic FC 500. The percentage of apoptotic cells was determined using an Annexin V-FITC/7-AAD kit (Beckman Coulter, USA). Harvested cells were suspended in binding buffer, stained with Annexin V-FITC and 7-AAD Viability Dye for 15 min, and analyzed within 30 min by flow cytometry. Cells with high levels of FITC-annexin V binding but low 7-AAD staining were considered as apoptotic cells.

## Figures and Tables

**Figure 1 ijms-24-15736-f001:**
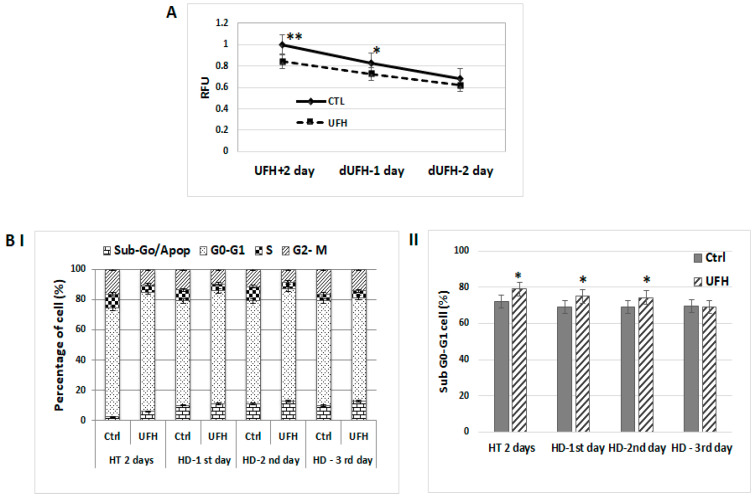
Anti-tumorigenic effects of heparin persist even after heparin has been discontinued in breast cancer cells. MCF-7 cells were grown in medium containing UFH (10 IU/mL) for two days, and then changed into fresh medium without heparin. Cultures were maintained and sampled on 1, 2, or 3 days for the following assays. (**A**) Cell proliferation assay showing that a reduced cell proliferation rate occurred in the presence of heparin (UFH) which persisted after heparin was discontinued (dUFH) in cultures maintained for a further 1 and 2 days. Asterisks indicate the degree of statistical significance (* *p* < 0.05; ** *p* < 0.01) as determined in comparison with the control untreated group. (**B**) Cell cycle assay. Cell cycle profiling of heparin-treated MCF-7 cells (HT) showed arrest in the G0-G1 phase and also a reduction in the S-phase in comparison to untreated control cells. These differences persisted on the first and second day after heparin treatment was discontinued (HD). (**I**) Cell-cycle distribution according to the percentage of the cell population in each phase. (**II**) The percentage population of the sub-G1 cell population shown in comparison to that of control cells. Data are representative of at least three independent experiments and shown as the mean ± SD (* *p* < 0.05).

**Figure 2 ijms-24-15736-f002:**
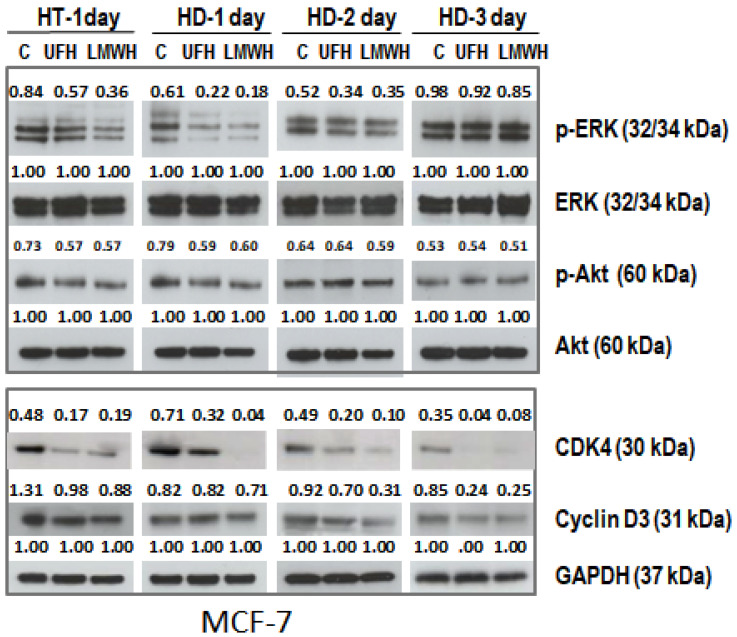
p-ERK and p-Akt and levels of cell cycle regulators, CDK 4 and cyclin-D3 were reduced in heparin (UFH and LMWH)-treated MCF-7 cells (HT) in comparison to untreated control cells. These differences persisted even after heparin had been discontinued (HD) in MCF-7 cells. The number above each band is the level of expression of p-ERK and p-AKT calculated with respect to the level of ERK or AKT, and the level of expression of CDK 4 and cyclin-D3 calculated with respect to GAPDH.

**Figure 3 ijms-24-15736-f003:**
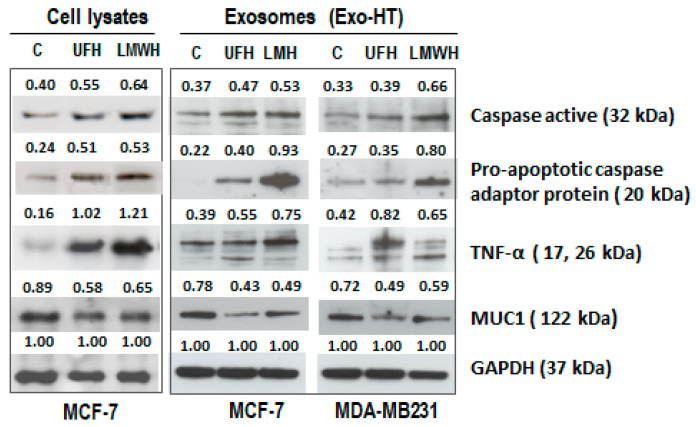
Heparin exerts a similar modulatory effect on pro-tumorigenic proteins within breast cancer cells and their sEV. Western blot assay was carried out on sEV separated from the culture medium of UFH and LMWH (10 IU/mL) treated MCF-7 and MDA-MB231 cells and, for comparison, whole cell lysates of MCF-7 cells. The results demonstrate that the heparin-induced modulation of the expression of pro-tumorigenic proteins in sEV in comparison to a control heparin-untreated sample of sEV (C) was similar to that observed in cell lysates from heparin-treated MCF-7 cells.

**Figure 4 ijms-24-15736-f004:**
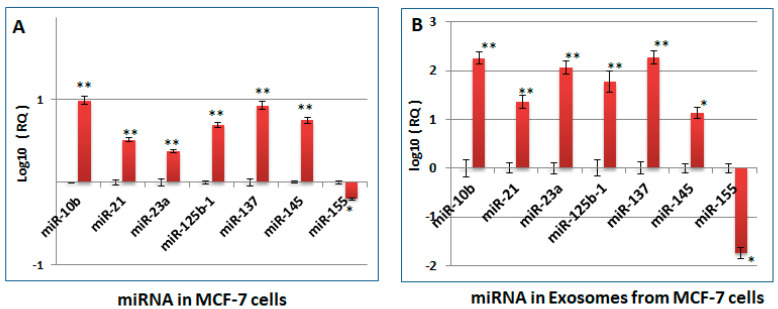
A selected miRNA profile of both MCF-7 cells and sEV was modulated upon treatment with heparin. (**A**) A selected miRNA expression profile as determined in RNA isolated from the heparin-treated whole cells; (**B**) A selected miRNA expression profile of RNA isolated from sEV derived from the heparin-treated cells. The expression profile in cells and sEV of untreated cells is used as control (log10 = 0). The results show a high degree of similarity between heparin-induced modulation of the profile when comparing that of the exosome and the parent cell. Asterisks indicate the degree of statistical significance (* *p* < 0.05; ** *p* < 0.01) as determined, respectively, in comparison to the control group.

**Figure 5 ijms-24-15736-f005:**
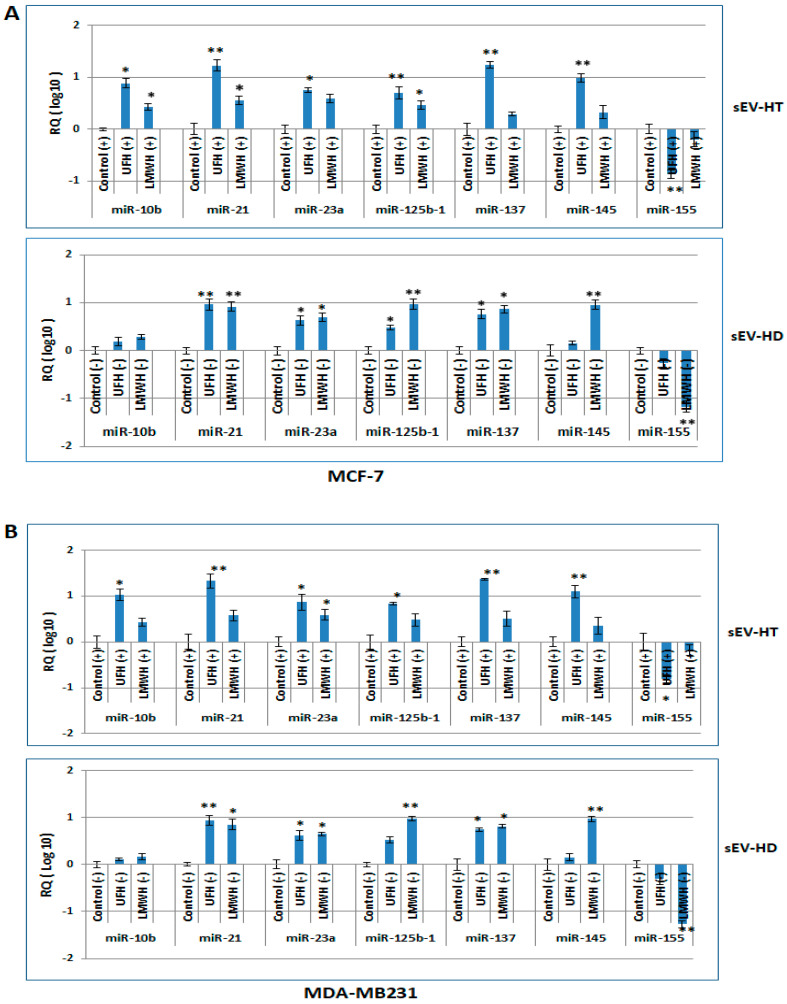
The selected miRNA profile of MCF-7 and MDA-MB231 cells was modulated by heparin-treatment-derived sEV. A selected miRNA profile was determined in RNA isolated from MCF-7 cells (**A**) and MDA-MB231 cells (**B**) which have been treated for 24 h with each of two types of sEV preparations (added at a final concentration of 50 µg/mL), namely, sEV prepared from cells undergoing treatment with either UFH or LMWH (10 IU/mL) (sEV-HT), and from cells in which heparin treatment had been discontinued (sEV-HD). A similar alteration of the miRNA expression profile was observed in cells treated with sEV-HT or sEV-HD in comparison to cells treated with sEV prepared from cells cultured in the absence of heparin (sEV-C). Asterisks indicate the degree of statistical significance (* *p* < 0.05; ** *p* < 0.01) as determined, respectively, in comparison with the control group.

**Figure 6 ijms-24-15736-f006:**
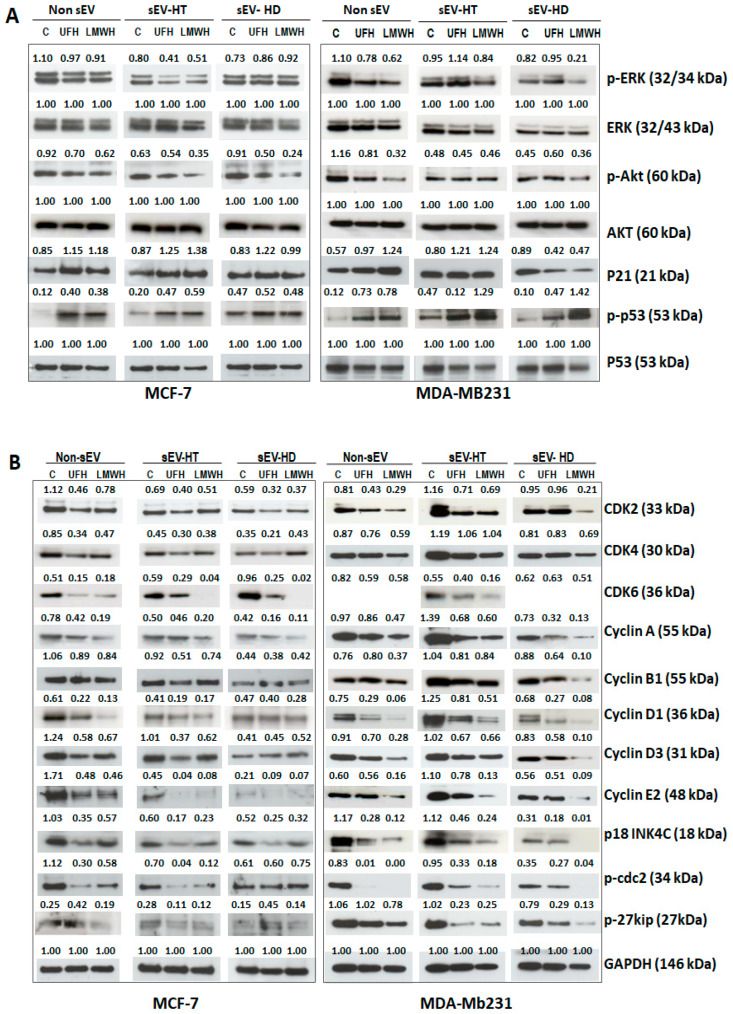
sEV from heparin-treated cells enable to modulate pro-tumorigenic and cell cycle regulatory proteins in cancer cells. The cells were treated for 24 h with each of two types of sEV preparations (added at a final concentration of 50 µg/mL), namely, sEV prepared from cells undergoing heparin treatment (sEV-HT) and sEV prepared from cells in which heparin treatment had been discontinued (sEV-HD), and the results were compared to those of cells treated with control sEV from heparin-untreated cells (sEV-C). Western blot was performed and compared to samples of cell lysate from heparin-treated cells (UFH or LWMH 10 IU/mL). (**A**) Heparin-treated sEV (sEV-HT) are able to modulate the expression of pro-tumorigenic proteins and signaling activities in MCF-7 and MDA-MB231 cells in a similar way to that caused by the direct treatment of cells with heparin (non-sEV). (**B**) Treatment of MCF-7 and MDA-MB231 cells with sEV-HT and sEV-HD downregulated the level of cell cycle regulatory marker proteins. The number above each band is the level of expression calculated with respect to the level of ERK, AKT, p53, or GAPDH.

**Figure 7 ijms-24-15736-f007:**
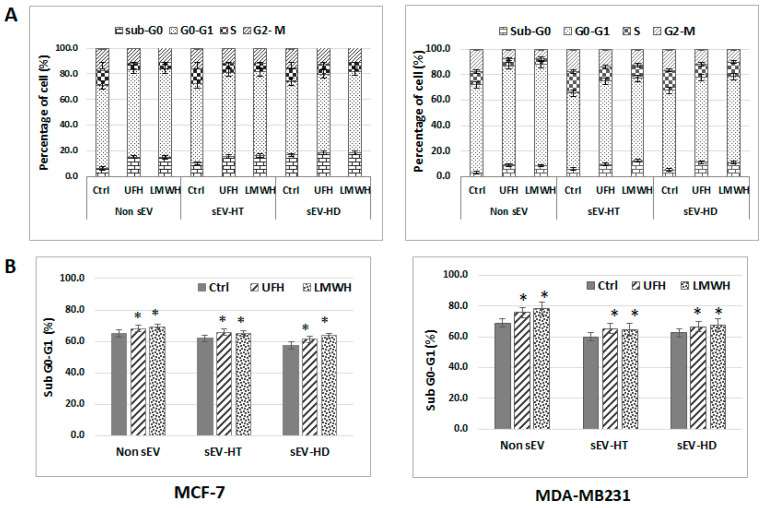
The cytological profile of breast cancer cells was modulated upon treatment with SEV-HT and SEV-HD. The cells were treated for 24 h with each of two types of sEV preparations (added at a final concentration of 50 µg/mL) namely sEV prepared from cells undergoing heparin treatment (sEV-HT) and from cells in which heparin treatment had been discontinued (sEV-HD). (**A**) The cell cycle assay showed an arrest in the G0-G1 phase, and a reduction in the S and G2-M phases in both the sEV-HT- and sEV-HD-treated group in a similar way to that caused by the direct treatment of cells with heparin (non-sEV). (**B**) The variation in the percentage of MCF-7 and MDA-MB231 cells in the sub G0-G1 population in different treatment groups. Data are representative of at least three independent experiments and shown as the mean ± SD (* *p* < 0.05; vs. control non-heparin-treated group).

**Figure 8 ijms-24-15736-f008:**
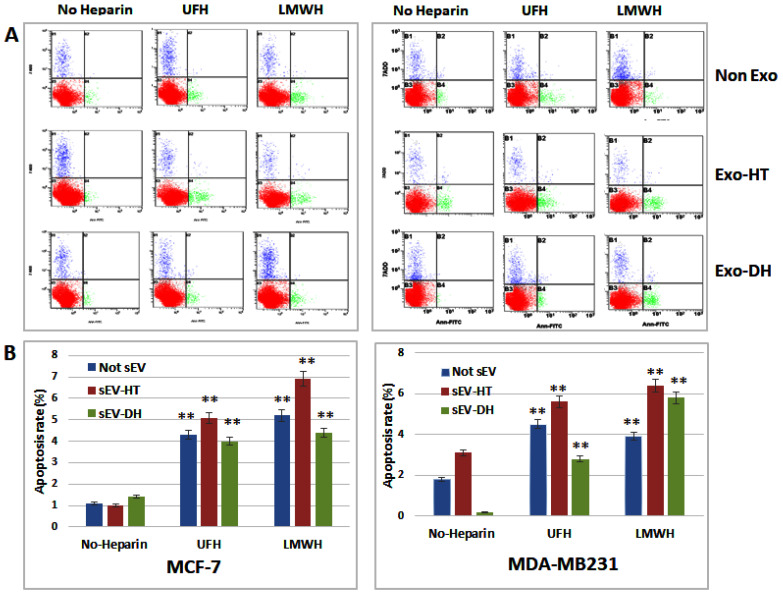
Increased apoptotic rate was observed in sEV-HT- and sEV-HD-treated groups, in which cell stained with annexin V-FITC and 7-AAD dye followed by flow cytometric analysis. Again, the modulation of the cell cycle and apoptosis profile were very similar to that caused by the direct treatment of cells with heparin (no sEV). (**A**) Graph panel represents from one of three representative experiments. Red color group represents viable cells, blue one is dead cells and green one is apoptotic cells. (**B**) The mean values of the apoptotic rate ± SD in different groups. ** *p* <0.01 vs. control non-heparin-treated group.

**Figure 9 ijms-24-15736-f009:**
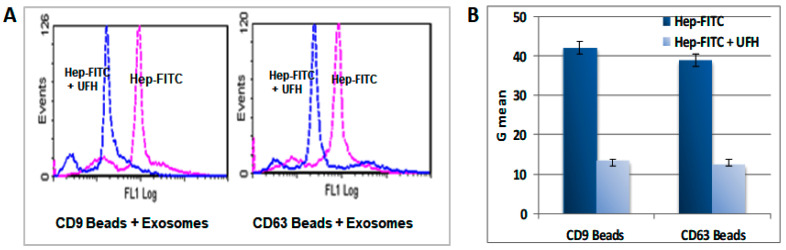
Flow cytometric analysis of heparin binding to sEV from MCF-7 cells. MCF-7 cells were treated with FITC-heparin (1 ug/mL) overnight. sEV were isolated from the culture medium of FITC-heparin-treated MCF-7 cells using an ExoQuick precipitation kit (System Biosciences, CA, USA). Analysis was carried out using an Exo-FLOW exosome kit in which magnetic beads (System Biosciences) pre-coupled with antibody to CD9 or CD64 (exosome surface markers), respectively, were used for the selective capture of two distinct sEV populations. The binding of FITC-heparin to the sEV was assayed by flow cytometry. (**A**) Histogram of population analysis by flow cytometry; (**B**) G means of flow cytometry data. The results are shown in the presence or absence of UFH (2 IU/mL) to demonstrate that binding is blocked by unlabeled heparin.

## Data Availability

The data that support the findings of this study are available from the corresponding author upon reasonable request.

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
