# Peer review of "The Tumorigenicity of Breast Cancer Cells Is Reduced upon Treatment with Small Extracellular Vesicles Isolated from Heparin Treated Cell Cultures"

_ijms, 2023, doi:10.3390/ijms242115736_

Round 1

Reviewer 1 Report

Comments and Suggestions for Authors

The authors of the ijms-2616365 manuscript claim that heparin has anti-tumorigenic effects on breast cancer cells in vitro, and hypothesize that small extracellular vesicles released by heparin-treated cells could be used to extend this effect over time. The manuscript is relatively well written, but I have some major concerns as follows that must be addressed before the manuscript can be considered for publication.

MAJOR

The authors say that “The cell proliferation rate was reduced in the heparin treated group and this reduction was maintained after the removal of heparin as observed on days 1 and 2 (Figure 1A)”. But according to Figure 1A, it doesn’t seem that a significant difference in cell proliferation was observed after discontinuing heparin treatment. The lines tend to get very close with time, and there is no statistical difference for any time points. Also, why haven’t the authors shown the 3 days time point in Figure 1A?

The authors claim that the treated MCF-7 cells showed a reduction of S-phase. Why don’t you show this result? Also, the authors claim that heparin has an anti-tumorigenic effect on breast cancer cells, so why does Figure 1 show only results for MCF-7, and then MDA cells are explored later as well?

According to the Minimal information for studies of extracellular vesicles 2018 (MISEV2018), any study using EVs should present a minimum characterization of the particles in other to validate the presence of EVs in the sample. To be considered for publication, authors should show at least data about EV size and concentration, electron microscopy images, and western blot of EV/EV contaminant markers.

Considering that for some experimental conditions, the cells were cultured for up to 5 days without FBS, it is very important to mention their % of viability at the moment of EV isolation. Could you please add this data for all experimental groups?

MINOR

Line 49: I believe the authors meant “sEV are known to act as intercellular messengers” instead of intracellular.

The miRNA paragraph in the introduction seems dismissible.

As Western blot is an important part of the study, please provide more detailed information for the methods, such as how cells were lysed, what method was used to quantify the proteins, how much protein was applied to the gels, conditions of run/transfer/probing, etc.

In the sEV isolation section of methods, please include the rotor used in the ultracentrifuge so that others can reproduce the protocol.

Why have the authors used a precipitation kit (which is known for altering the surface properties of EVs) to test whether FITC-heparin was capable of binding to the surface of sEV? Wouldn’t it be better to use the same UC-isolated ones?

What was used as a control (vehicle) for the biological experiments? Please describe it in the methods.

Comments on the Quality of English Language

Several small grammatical errors and misspelling should be revised. 

Author Response

1st Comments and Suggestions for Authors

The authors of the ijms-2616365 manuscript claim that heparin has anti-tumorigenic effects on breast cancer cells in vitro, and hypothesize that small extracellular vesicles released by heparin-treated cells could be used to extend this effect over time. The manuscript is relatively well written, but I have some major concerns as follows that must be addressed before the manuscript can be considered for publication.

MAJOR

The authors say that “The cell proliferation rate was reduced in the heparin treated group and this reduction was maintained after the removal of heparin as observed on days 1 and 2 (Figure 1A)”. But according to Figure 1A, it doesn’t seem that a significant difference in cell proliferation was observed after discontinuing heparin treatment. The lines tend to get very close with time, and there is no statistical difference for any time points. Also, why haven’t the authors shown the 3 days time point in Figure 1A?

R:  Heparin treatment of ther cell cultures was undertaken  under serum free conditions,- after  2 days the condition of the cells was  OK , but after 3 days, it began to deteriorate  The intention of the data shown in  Fig1A was merely to demonstrate that after UFH+2days, dUFH day1 (heparin discontinued) differences can still be seen , but after dUFH 2day, the proliferation rates were reduced under both sets of condition because of the  serum free culture . This also can concluded  from Fig1 B (II) - the cell cycle assay.     

The authors claim that the treated MCF-7 cells showed a reduction of S-phase. Why don’t you show this result? Also, the authors claim that heparin has an anti-tumorigenic effect on breast cancer cells, so why does Figure 1 show only results for MCF-7, and then MDA cells are explored later as well?

R: The figure 1 BI is a bar figure in which each phase of cell cycle was shown with different mark, in which can see G0-G1 and S phase exhibit difference between Ctrl and UFH treated group except in the HD- 3 day group. To address this, we only show the difference of G0-G1 phases (Fig 1B II)   as to represent,  all data for each phase list /add would tend to make  this figure be too complicated. We elected to follow the common way of some previous published papers with this type of cycle figure.

Our previous work had found that heparin treatment was able  to reduce the pro-tumorigenic profile , such as: adhesion, invasion and migration activity, and pro-tumorigenic genes and proteins as well in several human breast cancer cell lines, which included MCF-7 and MDA-MB (Chen et al., Thromb Haemost,  2013 109(6):1148-57),  so MDA-MB231 cells were used in this follow up of our previous  investigation. 

According to the Minimal information for studies of extracellular vesicles 2018 (MISEV2018), any study using EVs should present a minimum characterization of the particles in other to validate the presence of EVs in the sample. To be considered for publication, authors should show at least data about EV size and concentration, electron microscopy images, and western blot of EV/EV contaminant markers.

R: In Material and Methods, it has been mentioned that EVs were counted using qNano Gold (IZON Science Europe, UK). Based on the assay of the number count and concentration of EVs were used to following work. Regarding the size of EVs we used, qNano Gold assay which showed  that 85% were  between 50-150 nm, and  95% were  between 50 – 500 nm. Thank you for your suggestion which we have added this figure in manuscript.

In another hand,  our work via flow cytometric analysis of FITC-heparin treated sEV also shown the binding to Exo-Flow FACS magnetic beads pre-coated with sEV surface marker antibodies, CD9 or CD63. Such information should be sufficient  for minimum characterization of the  EVs used  in this investigation.   

MINOR

Line 49: I believe the authors meant “sEV are known to act as intercellular messengers” instead of intracellular.

R: This has been changed.

The miRNA paragraph in the introduction seems dismissible.

Agreed, it has been removed. 

As Western blot is an important part of the study, please provide more detailed information for the methods, such as how cells were lysed, what method was used to quantify the proteins, how much protein was applied to the gels, conditions of run/transfer/probing, etc.

R: Harvested cells were treated with lysis buffer (8M urea, 5%SDS, 40mM Tris-HCl pH6.8 and 1x Protease Inhibitor Cocktail tablet cOmplete (Roche, Germany)).  5 μg of the resulting protein sample was added per well and  run on SDS-PAGE Gel, and subjected to trans-blotting  with Trans-Blot turbo transfer system (Bio-Rad, UK), and resulting  blot was then analysed by iBright-CL1000 (Invitrogen).

In the sEV isolation section of methods, please include the rotor used in the ultracentrifuge so that others can reproduce the protocol.

R: It is Beckman XL90 Ultracentrifuge XL90, rotor type 70 TI, added 

Why have the authors used a precipitation kit (which is known for altering the surface properties of EVs) to test whether FITC-heparin was capable of binding to the surface of sEV? Wouldn’t it be better to use the same UC-isolated ones?

R: Your consideration is right, the precipitation may impact the surface precipitation of EVs, but at that time this kit was suggested as the best one to suitable for exosomes marker analysis (CD9/CD63)  and thereby to correlate with heparin binding.

Following are the manufacturers suggestions  about the uses of this  kit :

  • ExoQuickis a proprietary polymer that gently precipitates
  • Can be used to isolate exosomes for a wide range of downstream applications, including biomarker studies, exosomal miRNA profiling, exosomal proteomics, exosomal lipidomics/metabolomics, functional studies, such as in cell-to-cell signaling, and basic biology, such as roles in tumorigenesis.
  • In electron microscopy studies, exosomes isolated with ExoQuick appear similar to exosomes isolated using ultracentrifugation1-2, and these exosomes are also active in numerous functional assays.

What was used as a control (vehicle) for the biological experiments? Please describe it in the methods.

R: Added in method:

sEVs from untreated cell used as control   

Comments on the Quality of English Language

Several small grammatical errors and misspelling should be revised. 

R: have done

Reviewer 2 Report

Comments and Suggestions for Authors

Authors present a work addressing: ‘.The tumorigenicity of breast cancer cells is reduced upon treatment with small extracellular vesicles isolated from heparin-treated cell cultures’. Authors compared the properties of sEV prepared from heparin treated breast cancer cells (sEV-HT) to those from untreated cells (sEV-Ctrl) as well as to those separated from conditioned medium following the termination of treatment (heparin discontinued, sEV-HD). The general conclusion demonstrates that sEV may constitute a relatively stable reservoir of circulating heparin allowing heparin activity to persist in the circulation even after therapy has been discontinued. These findings can be considered as a special additional pharmacological characteristic of heparin clinical therapy. Thus I recommend publication after some mino issues have been addressed:

Please in discussion section add paragraph related to clinical and practical aspects of the study. How we can applicate your results into practice?, why your work is valuable in the field?
The authors should provide strengths and limitations of the study.

General: fantastic, well-conducted work.

Round 2

Reviewer 1 Report

Comments and Suggestions for Authors

Thank you for your replies. Please find my main comments bellow.

Although you say that differences can still be seen in Fig. 1A, they are not statistically significant differences. This is a very important result for the paper as it is the starting point for the story that heparin has an antitumorigenic effect, and this can be maintained after treatment discontinuation. The results do not allow this conclusion. 

It is still unclear why the authors show results for MDA cells in only part of the results.  

I could not find the qNano Gold results in the manuscript. 

True that you show minimum EV characterization, but it is very minimal, which does not confirm that EVs are intact structures and free of contamination. The results you show could be induced by damaged EVs, protein aggregate contaminants, etc. It is not enough for a study on the functional effects of EVs. 

Comments on the Quality of English Language

N/A

Author Response

Thank you very much for taking the time to review this manuscript again. Please find our detailed responses below . The corresponding revisions/corrections highlighted in the re-submitted files.

Reviewer 1’s Report 2

Thank you for your replies. Please find our main comments below.

Although you say that differences can still be seen in Fig. 1A, they are not statistically significant differences. This is a very important result for the paper as it is the starting point for the story that heparin has an antitumorigenic effect, and this can be maintained after treatment discontinuation. The results do not allow this conclusion. 

The original Fig 1A submitted was   in colour and the statistical difference markers disappeared when it was changed to Black & white one during editing. The statistically significant difference was referred to in the legends (Fig1A).  The missed statistically differences marker, has now been back to Fig 1A. Thanks for you for pointing out the discrepancy.  

It is still unclear why the authors show results for MDA cells in only part of the results.  

As we described our previous work had found that heparin treatment was able to reduce the pro-tumorigenic profile of several human breast cancer cell lines, which included MCF-7 and MDA-MB (Chen et al., Thromb Haemost, 2013 109(6):1148-57),  so MDA-MB231 cells were used as part of the introduction  in this follow up of our previous  investigation. 

I could not find the qNano Gold results in the manuscript. 

The qNano Gold results are highlighted in lines 118-122:  

Isolated sEVs were also counted and assayed using qNano Gold (IZON Science Europe, UK), which showed that 85% of them were between 50-150 nm, and 95% of them were between 50 – 500 nm.  Based on the assay of the number count and concentration of EVs, breast cancer cells were treated with 5-10 µg of sEV protein concentration /ml or 4 x 109 sEV / ml in the  work.

True that you show minimum EV characterization, but it is very minimal, which does not confirm that EVs are intact structures and free of contamination. The results you show could be induced by damaged EVs, protein aggregate contaminants, etc. It is not enough for a study on the functional effects of EVs.

sEV isolation was carried out using the procedure used in numerous current studies (Reference shown below) . The point the reviewer makes has validity but the widespread use of this technique demonstrates that it is acceptable for the study of the biology and function of  sEVs.

Reference:

  1. LeBleu, R.K.a.V.S., The biology, function, and biomedical applications of exosomes. Science. 2020, 367(6478).
  2. Peng ZY, et al. Downregulation of exosome-encapsulated miR-548c-5p is associated with poor prognosis in colorectal cancer. J Cell Biochem. 2018, 1-7.
  3. Vermes I, et al. Flow cytometry of apoptotic cell death. Journal of Immunological Methods. 2000, 243(1): 167-190.
  4. ROY J. CARVER. BIOTECHNOLOGY CENTER https://biotech.illinois.edu/flowcytometry/protocols/cell-cycle-analysis
  5. Wang S, et al. Role of exosomes in hepatocellular carcinoma cell mobility alteration. Oncol Lett. 2017, 14(6): 8122-8131.
  6. Rajput A et al., Exosomesas New Generation Vehicles for Drug Delivery: Biomedical Applications and Future Perspectives. Molecules. 2022 27(21): 7289. 
  7. Bowers EC et al., In vitro Models of Exosome Biology and Toxicology: New Frontiers in Biomedical Research. Toxicol In Vitro. 2020 64: 104462.
  8. Rahbarghazi R et al., Tumor-derived extracellular vesicles: reliable tools for Cancer diagnosis and clinical applications. Cell Commun Signal. 2019 17(1):73.

Round 3

Reviewer 1 Report

Comments and Suggestions for Authors

 Thank you for your responses. 

Comments on the Quality of English Language

N/A

Author Response

Thank you very much again for taking the time to review this manuscript. Please find our detailed responses below. The corresponding revisions/corrections were highlighted in the re-submitted files.

After reading the manuscript and the authors’ responses to the reviewers’ comments, I recommend that the authors do some additional work on the text of the paper.

 1) Fig. 1 does nʹt contain any C...

please see Line 172, Fig. 1C

 2) Fig. 2 has not been mentioned in the text

 Both suffering from the same mistake, we are sorry about this. In both case , it refers to Fig 2 , not Fig 1. This error has now been corrected    

3) General recommendation for Figures, composed of two distinct images positioned on separate pages. They should be named not Fig. 6A and Fig. 6B, but Fig. 6 and Fig. 7, correspondingly. The same is relevant in case Fig. 7, and Fig. 5

 We consider that Fig 6A and B should be published side by side since they both have exactly the same legend as will be sorted out in the final edit. Exactly the same consideration applied to fig 5.  

Fig 7 has divided as two figures (Figure 7 and 8)

4) The manuscript and Figures contain multiple typos, for example, what one can see in Fig.7: 

Non Heparin, No-Heparin; NonsEV, Non sEV

 These errors have been corrected

5) BUT THE MAJOR CONCERN is the correspondence of the conclusions to the results obtained (as one of the reviewers wrote about).

5.1) The authors write in the Discussion: ʺThe significance of our findings in terms of the clinical use of heparin during cancer treatment is that it provides an additional mechanism by which heparin and heparin like molecules are able to exert a favourable effect on the long term prognosis of cancer patients.ʺ

 Which evidence provides the results in the context of this sentence?

 The following has now been included in the corrected manuscript

It is well known that heparin, including UFH, low-molecular-weight heparin (LMWH) and heparin derivatives, are commonly used in venous thromboembolism treatment and reportedly have beneficial effects on cancer survival (43) There are number of reports shown that LMWH can improve the 3-month and 6-month survival of cancer patients (44-49).  For example, a multicentre clinical trial exploring the influence of anticoagulant treatment in 277 small cell lung cancer patients showed that 5-week subcutaneous heparin treatment led to substantially improved survival rates compared to no treatment at 1, 2 and 3 years (40% vs. 30%, 11% vs. 9%, and 9% vs. 6%, respectively) (50). Another clinical study found that death rates in ovarian cancer patients at a 2-year postoperative follow-up were 24% following treatment with certoparin compared to 37.5% following treatment with UFH, suggesting that LMWHs are better than UFN in improving survival rates (51)

Reference

43. Ma SN , Mao ZX,  Wu Y, et al., The anti-cancer properties of heparin and its derivatives: a review and prospect. Cell Adh Migr. 2020; 14(1): 118–128
  1. Icli F, Akbulut H, Utkan G, et al. Low molecular weight heparin (LMWH) increases the efficacy of cisplatinum plus gemcitabine combination in advanced pancreatic cancer. J Surg Oncol. 2007; 95:507–512.
  2. Prandoni P, Lensing AW, Buller HR, et al. Comparison of subcutaneous low-molecular-weight heparin with intravenous standard heparin in proximal deep-vein thrombosis. Lancet. 1992; 339:441–445.
  3. Gould MK, Dembitzer AD, Doyle RL, et al. Low-molecular-weight heparins compared with unfractionated heparin for treatment of acute deep venous thrombosis. A meta-analysis of randomized, controlled trials. Ann Intern Med. 1999; 130:800–809.

47.Yoshitomi Y, Nakanishi H, Kusano Y, et al. Inhibition of experimental lung metastases of Lewis lung carcinoma cells by chemically modified heparin with reduced anticoagulant activity. Cancer Lett. 2004; 207:165–174.

  1. Borsig L, Wong R, Hynes RO, et al. Synergistic effects of L- and P-selectin in facilitating tumor metastasis can involve non-mucin ligands and implicate leukocytes as enhancers of metastasis. Proc Natl Acad Sci U S A. 2002; 99:2193–2198. 
  2. Green D, Hull RD, Brant R, et al. Lower mortality in cancer patients treated with low-molecular-weight versus standard heparin. Lancet. 1992; 339:1476. 
  3. Lebeau B, Chastang C, Brechot JM, et al. Subcutaneous heparin treatment increases survival in small cell lung cancer. “Petites Cellules” Group. Cancer. 1994; 74:38–45.
  4. Von Tempelhoff GF, Harenberg J, Niemann F, et al. Effect of low molecular weight heparin (Certoparin) versus unfractionated heparin on cancer survival following breast and pelvic cancer surgery: A prospective randomized double-blind trial. Int J Oncol. 2000; 16:815–824.

5.2) ʺOur findings are the first to establish that heparin can modulate the progression of cancer via the intracellular messenger role of exosomes released by cancer cells turning their role to an antitumorigenic one, rather than a protumorigenic oneʺ   

Exosomes of sEV?

Sorry it should be changes due to sEVs

5.3) What is the physiological mechanism by which treatment with heparin has such an effect on exosomes?

 As described in the Introduction and Discussion, our previous work established an anti-tumorigenic effect of heparin treatment of cancer cells as observed in changes to the genotype and phenotype of cells cultured in the presence of heparin . This present study establishes that these changes in  sEVs produced by the heparin treated cells.  sEV are remarkably stable in bodily fluids, and able to interact with the surrounding tissues in both intercellular and extracellular domains, the sEV heparin may also constitute a relatively stable reservoir of circulating heparin allowing heparin activity to persist in the circulation even after therapy has been discontinued.  It is a special effect of heparin-derived sEV for heparin treatment .

 Which exosomal components are changed due to the heparin treatment (lipids, proteins, nucleic acids)?

Our work as is shown in  Figure 3 observed  that hHeparin exerts a similar modulatory effect on pro-tumorigenic proteins within the breast cancer cells and their sEVs.

Do these changes occur on the surface of exosomes? Answers to these questions should be added.

We have as yet to investigate whether the observed changes in protein expression occur on the surface of sEV or within and further work is planned. This sentence is now included within the Discussion  
